# Performative Prediction on Games and Mechanism Design

**António Góis** [1]  **Mehrnaz Mofakhami** [1]  **Fernando P. Santos** [2]  **Simon Lacoste-Julien** [1][3]  **Gauthier Gidel** [1][3]

## Abstract

Predictions often influence the reality which they aim to predict, an effect known as performativity. Existing work focuses on accuracy maximization under this effect, but model deployment may have important unintended impacts, especially in multi-agent scenarios. In this work, we investigate performative prediction in a concrete game-theoretic setting where social welfare is an alternative objective to accuracy maximization. We explore a collective risk dilemma scenario where maximising accuracy can negatively impact social welfare, when predicting collective behaviours. By assuming knowledge of a Bayesian agent behavior model, we then show how to achieve better trade-offs and use them for mechanism design.

## 1. Introduction

Recent frameworks such as performative prediction study how predictions influence the distribution they aim to predict (Hardt & Mendler-Dünner, 2023). These have focused on accuracy for one predictor and independent predicted agents: a spam producer changes its content to fool a spam classifier (Dalvi et al., 2004; Hardt et al., 2016), or one loan applicant adapts to improve its credit score ignoring adaptation by others (Perdomo et al., 2020).

Performative prediction typically considers a larger set of independent data points, but interdependencies among predicted agents are not explicitly modeled. However a plethora of examples exists requiring a collective scale. Financial markets are filled with self-fulfilling prophecies (Soros, 1987). These may have actually deeply harmed society in cases such as the British pound collapse in 1992 (Naef, 2022), highlighting that accuracy is not the only metric of concern. Examples of interdependent populations abound, with implications on social welfare, such as road traffic prediction, policy-making to handle the risk of pandemics or climate change, and even election polls (Simon, 1954).

Multi-agent extensions of performative prediction have focused mostly on multiple predictors (Li et al., 2022; Piliouras & Yu, 2022; Narang et al., 2022; Wang et al., 2023). In Eilat et al. (2022) prediction outcomes depend on a graph $\mathcal{G}$ because the classifier assumes it. Mendler-Dünner et al. (2022) mention spill-over effects as a way to give a causal treatment to social influence. Hardt et al. (2023) consider predicted agents that coordinate to influence the training of a classifier. To model a dilemma such as cooperation for climate change, here we propose the first setting with inherent interdependence among predicted agents. Agents play a cooperation game whose outcome depends locally on others' actions, and decisions are influenced by predictions.

Additionally, the broad goal of existing frameworks has been to maximize accuracy under performativity (Miller et al., 2021). However, accuracy is not necessarily the only goal of predictions. These can be used as part of mechanism design, particularly in interdependent settings. Recommender systems may wish to preserve content diversity (Eilat & Rosenfeld, 2023). Vo et al. (2024) consider a trade-off between selecting good candidates and maximizing their improvement, with consequences for agent welfare. In collective scenarios, predictions of pandemic growth or climate change can inform public policy, and become performative if risk is successfully reduced. In financial markets, predictions may aim at maximizing profit instead of accuracy. In elections, each candidate wishes to push the forecast whose collective reaction will benefit them the most. Even if a neutral entity wishes to deploy an accurate election poll, its performative effect may have strong unintended consequences (Blais et al., 2006; Westwood et al., 2020; Nina et al., 2023).

While deliberately deploying a wrong prediction is not an ethical option, there may be multiple possible realities that can be induced (Hardt et al., 2022) — therefore different predictions may be equally correct. Providing a snapshot of pre-prediction reality may be a way out of this dilemma, but can be wrongly interpreted as a prediction of post-prediction reality. The choice of how many snapshots to provide before action will also influence arbitrarily the outcome. Our work illustrates this problem and difficult choices that arise from it, through the following contributions:

- We propose the first performative setting where the

[1] Mila, Université de Montréal [2] Informatics Institute, University of Amsterdam [3] Canada CIFAR AI Chair. Correspondence to: António Góis .

*Accepted to ICML 2024 Workshop on Agentic Markets*, Vienna, Austria. Copyright 2024 by the author(s).

predicted population is inherently interdependent.

- We use predictions as a mechanism to maximize social welfare, and explore trade-offs with accuracy.

## 2. A Model for Predicting Collective Action

We are interested in game-theoretic scenarios where a population is interdependent and possibly influenced by predictions of collective behaviour. This is motivated by multiple examples where individual outcomes depend on a group's action — adherence to measures for controlling a pandemic spread, protecting climate or governing common goods, among others.

To that end, we propose a model where subgroups from a larger population interact simultaneously in a given round, drawing inspiration from evolutionary game theory on networks (Smith, 1982; Ohtsuki et al., 2006). Given a graph $\mathcal{G} = (V, E)$, for any agent $i$, its group is composed of $i$ (itself) and its neighbors in the graph $\mathcal{N}(i)$. For one round of the game, agents simultaneously select an action, and each agent $i$ receives a payoff $\pi_i(a_i, a_{\mathcal{N}(i)})$ depending on its own action $a_i$ and on neighbours' $a_{\mathcal{N}(i)}$. The game repeats indefinitely.

To choose $\pi_i$, we focus on a game coined Collective Risk Dilemma (CRD; Milinski et al., 2008; Santos & Pacheco, 2011), suitable to study mechanism design (Góis et al., 2019). Each round requires a critical mass of cooperators to achieve success and prevent collective losses. This may represent the protection of common natural resources, the immunity of a partially vaccinated group, and the collective development of tools like Wikipedia or Linux, among many others. If the fraction of cooperators remains below a threshold $T$, everyone risks losing their endowment with probability $r$. Each agent chooses whether to defect ($a_i = 0$) or cooperate at a cost ($a_i = 1$), with payoffs described below:

**Definition 2.1.** (Defector's payoff) Let $\mathbb{1}[\cdot]$ be the indicator function. $k_i = \sum_{j \in \mathcal{N}(i) \cup \{i\}} a_j$ is the number of cooperators in agent $i$'s group. Given initial endowment $B$, $k_i$ cooperators in a group of size $M_i$, threshold $T$ where $0 \leq T \leq 1$, and risk $r$, where $0 \leq r \leq 1$, the payoff of defector $i$ is

$$\pi_{D_i}(k_i) = B \cdot (\mathbb{1}[k_i \geq \lceil TM_i \rceil] + (1-r)\mathbb{1}[k_i < \lceil TM_i \rceil])$$ (1)

**Definition 2.2.** (Cooperator's payoff) Given a cost $cB$ of cooperating, where $0 \leq c \leq 1$, the payoff of cooperator $i$ is

$$\pi_{C_i}(k_i) = \pi_{D_i}(k_i) - cB$$ (2)

A CRD is used as payoff function $\pi$ for all agents, using the same threshold value $T$ and unique $M_i$'s given by $\mathcal{G}$. This leads to partially aligned incentives — each agent $i$ gains

from preventing a disaster where $\frac{k_i}{M_i} < T$, but would rather avoid incurring cost $c$ of cooperating to increase $k_i$.

For one round of CRD with $c < r$ and one single group (where $\mathcal{G}$ is a clique $C$), the Nash equilibria are for everyone to defect (sub-optimal) or to have exactly $\lceil TM_i \rceil$ cooperators (Pareto optimal). The challenge is in coordinating a group towards the Pareto optimal Nash, which doesn't happen spontaneously in the real world (Milinski et al., 2008).

### 2.1. Agent Model

We model agents as computing a best-response, given expectations of other individuals' actions. To nudge behaviour, a predictor provides predictions of the population actions. Alternatively to correlated equilibria (Aumann, 1974) we provide a public signal, which agents can choose to trust or not. Since this signal is learned from global observations of the whole population (and not just locally) it has the potential to bring additional information to agents. We assume agents observe a public prediction of others' actions, but stop trusting it if it is inaccurate. More specifically, they follow a Bayesian update to compute the probability of trusting the prediction. Agent $i$ has two competing explanations for each neighbour $j$'s behaviour — the external prediction $\hat{\theta}_j$ and an internal expectation $\alpha_{i,j}$. Both $\hat{\theta}_j$ and $\alpha_{i,j}$ are Bernoulli parameters that estimate a hypothetical true parameter $\theta_j = P(a_j = 1)$. The probability $\tau_{t,i}$ of $i$ trusting the external predictor in timestep $t$ is given by:

$$\tau_{t,i} = \mathbb{P}(\text{trust}|\mathbf{a}, \hat{\theta}) = \frac{\tau_{t-1,i}\mathrm{P}_i(\hat{\theta}_t, \mathbf{a}_t)}{\tau_{t-1,i}\mathrm{P}_i(\hat{\theta}_t, \mathbf{a}_t) + (1-\tau_{t-1,i})\mathrm{P}_i(\boldsymbol{\alpha}_i, \mathbf{a}_t)}$$ (3)

with $\mathrm{P}_i(\boldsymbol{\theta}_t, \mathbf{a}_t) := \prod_{j \in \mathcal{N}(i)} \theta_{j,t}^{a_{j,t}} (1 - \theta_{j,t})^{1-a_{j,t}}$.

Given the expectation of others' actions, $i$ can compute a rational utility-maximizing action. As long as $c < r$, it is rational for $i$ to cooperate if and only if $\sum_{j \in \mathcal{N}(i)} a_j = \lceil TM \rceil - 1$. In words, $i$ cooperates when it is the only missing cooperator required to overcome the threshold in its group. Given probability $\boldsymbol{\theta}_{\mathcal{N}(i)} = \theta_{1...M_i-1}$ of each neighbour of $i$ to cooperate, a Poisson binomial distribution $g(\boldsymbol{\theta}_{\mathcal{N}(i)})$ gives us the aggregate probability of having $\lceil TM \rceil - 1$ cooperators in the group. Best-response becomes $\arg\max_{a_i} \mathbb{E}_{a_{\mathcal{N}(i)} \sim g(\boldsymbol{\theta}_{\mathcal{N}(i)})}[\pi(a_i, a_{\mathcal{N}(i)})]$. Then, $i$ cooperates if $r(\tau_i g(\hat{\theta}_{j \in \mathcal{N}(i)}) + (1 - \tau_i)g(\alpha_{i,j \in \mathcal{N}(i)})) > c$, and defects otherwise (Appendix A).

## 3. Model dynamics

### 3.1. Simple Environments

We begin by analyzing the following simplified setting:

**Assumption 3.1.** (Simple controllable setting) a) agents are

initialized with prior $\tau_0 = 1$ ignoring their internal beliefs $\alpha$, and b) predictions are binary: $\hat{\theta}_t \in \{0,1\}^{|V|}$.

Let a *self-fulfilling prophecy* be when $\forall i, a_i = \hat{\theta}_i$. Assuming binary predictions is useful in this definition, since $a$'s need to match $\theta$'s. Removing the interference of internal expectations $\alpha$ by having $\tau_0 = 1$, predictions become static: $\hat{\theta}_t = \hat{\theta}$. With full trust guaranteed, there is no need to balance between trust and other goals through time. Under Assumption 3.1, if agents are never indifferent between actions, predicting a strict Nash equilibrium is sufficient and necessary to have a self-fulfilling prophecy (i.e. $\forall i, \text{BestResponse}(\hat{\theta}_{\mathcal{N}(i)}) = \hat{\theta}_i$).

Whether there is a Nash equilibrium that maximizes social welfare determines whether the predictor must compromise accuracy to maximize it. Note that, as long as $\forall i, \lceil TM_i \rceil > 1$, *all-defecting* is always a self-fulfilling prophecy. This explains why accuracy maximizers empirically induce low-cooperation states. Using Assumption 3.1, the topology of $\mathcal{G}$ and threshold $T$ become the only constraints determining whether a given system state is attainable.

**Theorem 3.2.** *(Sufficient conditions for success) Let "full success" be the setting where $\forall i, \frac{k_i}{M_i} \geq T$. Under Assumption 3.1 and $c < r$, each of the following is a sufficient condition to have $\exists \hat{\theta} \implies$ full success, where $\hat{\theta}$ is a self-fulfilling prophecy:*

1. *$\mathcal{G} = C$, where $C$ is a clique or a fully connected graph: Assume $\hat{\theta}$ predicts a configuration with $k_i = \lceil TM_i \rceil$. Since all agents share the same group, it is not possible for one agent to deviate from $\hat{\theta}$ without lowering its $\pi$;*

2. *T=1: no agent can free-ride, since all are required to cooperate;*

3. *T=0: full success is guaranteed by default.*

Figure 1 illustrates the previous remarks, over a 3-node clique. a) and c) are Nash equilibria and self-fulfilling prophecies, while b) and d) are self-defeating prophecies. An accuracy maximizer would choose a) or c), while a welfare maximizer would choose a) or b). Here it is possible to maximize both quantities through a).

However, both goals may be at odds in other settings. In Figure 2 there is no prediction that satisfies simultaneously an accuracy maximizer and a welfare maximizer. The only self-fulfilling prophecy is a), reaching full-defection. Only e) reaches full success, but since it is not a Nash it is not self-fulfilling. This is because the center node could have achieved success while defecting, but this would have prevented success in groups of size 2. Note that in general full success is not always achievable, even if we do not require a self-fulfilling prophecy (Appendix F).

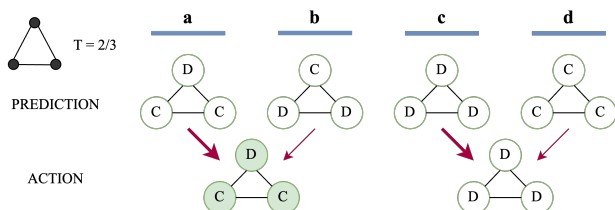

*Figure 1.* Dark nodes have achieved success, and thick arrows are self-fulfilling prophecies. Both a) and c) are self-fulfilling prophecies where accuracy is maximized, therefore an accuracy maximizer is indifferent between them. However, in a) full success is achieved, but in c) all fail. b) also maximizes group success but at the expense of 0% accuracy.

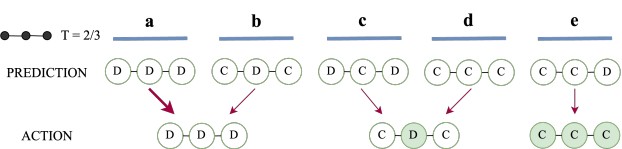

*Figure 2.* Dark nodes have achieved success, and thick arrows are self-fulfilling prophecies. Here there is no self-fulfilling prophecy which maximizes group success, forcing a trade-off between accuracy and group success. Only e) maximizes group success, but the center node regrets having cooperated. Note that, with $T = \frac{2}{3}$, groups of size $M_i = 2$ require both agents to cooperate.

This shows how different predictions induce different realities in this model. As a consequence, seeking only high-accuracy predictions may inadvertently induce low-cooperation states. The next section will further showcase this in richer environments, lifting the simplifying assumptions in 3.1.

### 3.2. Learned Predictor and Simulations

As the population size $|V|$ grows and internal expectations are allowed to differ from predictions ($\hat{\theta}_t \neq \alpha$), analysis becomes more complex. We resort to simulations and learned predictors to study larger systems.

We choose to represent the predictor through a neural network, which receives as input an embedding of the population's actions in the previous time-step: $\hat{\theta}_t = f_\phi(a_{t-1}) : \{0,1,t_0\}^{|V|} \to [0,1]^{|V|}$. The loss is either cross-entropy, a differentiable proxy for number of successful groups, or a combination of both following Sener & Koltun (2018). Each metric is the sum of 20 time-steps of a CRD. Gradients are computed assuming access to the inner behaviour of agents. To maximize the number of successful groups, it backpropagates through a differentiable version of their decision rule and of the payoff, where both step-functions

are replaced by sigmoids (Appendix B).

When optimizing for social welfare, the predictor still needs to consider prediction accuracy in order to maintain agents' trust. Let $\tilde{a}_{i,t} = \sigma(\pi_{C_i,t} - \pi_{D_i,t})$ be a differentiable proxy of agents' true decision rule $a_{i,t} = \mathbb{1}[\pi_{C_i,t} - \pi_{D_i,t} > 0]$. We analyze here a proxy goal $\hat{U}_C = \sum_{t=1}^{T}\sum_{i=1}^{N}\tilde{a}_{i,t}$ whose gradient can be decomposed in two components:

$$
\nabla_\phi \hat{U}_C = \sum_{t=1}^{T}\sum_{i=1}^{N}\psi_{t,i}(\phi)[
$$
$$
(g(\hat{\theta}_{j\in\mathcal{N}(i)}(\phi)) - g(\alpha_{i,j\in\mathcal{N}(i)}))\underbrace{\nabla_\phi\tau_{t,i}(\phi)}_{\text{accuracy}}
$$
$$
+ \tau_{t,i}(\phi)\underbrace{\nabla_\phi g(\hat{\theta}_{j\in\mathcal{N}(i)}(\phi))}_{\text{steering}})] \quad (4)
$$

$\psi_{t,i}(\phi) = \tilde{a}_{t,i}(1 - \tilde{a}_{t,i})rB$ is a scalar which is higher when agents are closer to flipping their choice of action between cooperation and defection. An optimizer using this goal needs to control accuracy to keep trust high, and steer towards higher cooperation when trust is high. In practice we use a slightly more complex goal $\hat{U}_{Pop}$ that is closer to true social welfare, leading to qualitatively similar empirical results and amenable to a similar analysis (Appendix B).

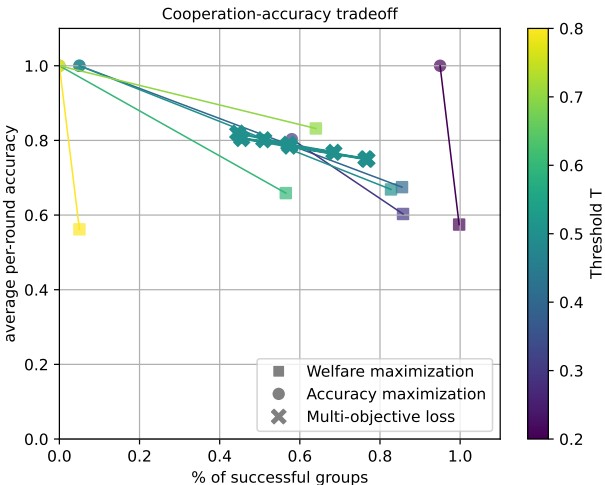

*Figure 3.* Accuracy vs. social welfare trade-off for different threshold values. Pareto front computed through multi-objective optimization for $T = 0.5$. All experiments were conducted using a scale-free $\mathcal{G}$ with 20 nodes and mean degree of 2 (Barabási & Albert, 1999), $c = 0.2, B = 1, r = 0.4, \alpha_{i,j} = 0.8$ and $\tau_0 = 0.5$.

In Figure 3 we observe the result of training either for accuracy or welfare maximization, for different values of threshold. Unless the threshold is very low ($T \in \{0.2, 0.3\}$), a predictor maximizing accuracy will induce states of very

low cooperation (find a related discussion in Appendix F). A predictor maximizing welfare can prevent this, but at the expense of accuracy. This is in line with § 3.1, where both metrics may be impossible to maximize simultaneously. To overcome this, we follow Sener & Koltun (2018) to jointly optimize for both metrics. For $T = 0.5$, we compute the Pareto front in this way.

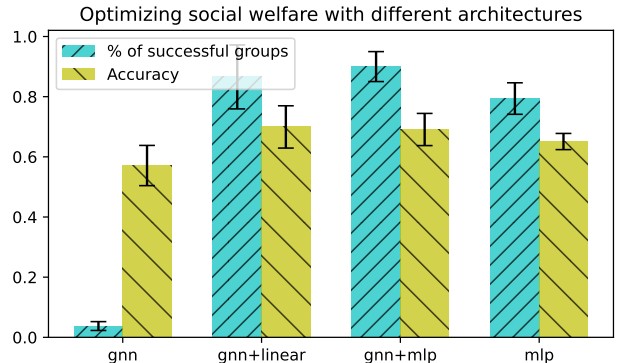

*Figure 4.* Performance of different architectures, when optimized to maximize social welfare. $T = 0.5$ and other parameters follow Figure 3.

Regarding architecture choices, we use a multilayer perceptron (MLP), a graph neural network (GNN), GNN+MLP and GNN+linear (Figure 4). For an MLP, a concatenation of all nodes' actions is provided as input, and their actions for the next step are jointly predicted. Having a GNN followed by an MLP or a linear layer provides a gain over MLP alone, by adding information about $\mathcal{G}$. Interestingly, GNNs alone, being the only model unable to do centralized coordination, are not able to promote cooperation. For two equal nodes, some settings may require one to cooperate and the other to defect. A GNN however is unable to provide different outputs to each node. When optimizing for accuracy, this limitation of GNNs goes by unnoticed (Appendix C).

## 4. Conclusion

We have introduced a framework to study performative effects under game-theoretic settings on a network of agents. We show how social welfare and accuracy can be in conflict, and empirically compute their Pareto front. Although accuracy may seem like a way to avoid manipulating reality, multiple accurate outcomes with different social welfare can be induced when performativity is strong enough. Ignoring side-effects may be more harmful than considering them, making it inevitable to regard performative prediction (partly) as mechanism design in our examples. It is important to connect this kind of model to real data in future work, despite challenges of doing so in performative settigs. We also plan to further develop theory and study other models of agent behaviour.

## 5. Acknowledgements

This research was partially supported by the Canada CIFAR AI Chair Program, by a grant from Samsung Electronics Co., Ldt., by an unrestricted gift from Google, and by a discovery grant from the Natural Sciences and Engineering Research Council of Canada (NSERC). F. P. Santos acknowledges funding by the European Union (ERC, RE-LINK, 101116987). Simon Lacoste-Julien is a CIFAR Associate Fellow in the Learning in Machines & Brains program.

We would like to thank Jose Gallego-Posada for the insightful comments and discussion during the development of this work, leading to the analyses in Section § 3.1.

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

## A. Best response

An agent's best response selects the action with highest expected payoff, between cooperation and defection. Let $k_i' = \sum_{j \in \mathcal{N}(i)} a_j$ be the number of cooperators in $i$'s group, excluding $i$ itself. The payoff gain of switching from defection to cooperation is

$$\pi_{C_i}(k_i' + 1) - \pi_{D_i}(k_i') = \begin{cases} (r-c)B & \text{if } k_i' = \lceil TM_i \rceil - 1 \\ -cB & \text{otherwise} \end{cases}$$

In words, $i$ gains $(r-c)B$ from cooperating when it is the last member required to overcome the threshold in its group. It loses $cB$ for any other group configuration. Its best response is then to cooperate when it is "at the threshold" ($k_i' = \lceil TM_i \rceil - 1$) and defect otherwise, as long as $c < r$.

Its expectation of others' actions depends on two competing explanations $g(\hat{\theta}_{j \in \mathcal{N}(i)})$ and $g(\alpha_{i,j \in \mathcal{N}(i)})$, and the likelihood $\tau_i$ of trusting the first option. Each explanation provides the likelihood $\mathbb{P}(k_i' = \lceil TM_i \rceil - 1) = g(\cdot)$, by using a Poisson binomial distribution to aggregate individual likelihoods of each neighbour to cooperate. It should then cooperate if

$$\mathbb{E}_{\tau_i}[\mathbb{E}_{g(\hat{\theta}_{j \in \mathcal{N}(i)})}[\mathbb{E}_{g(\alpha_{i,j \in \mathcal{N}(i)})}[\pi_{C_i}(k_i' + 1) - \pi_{D_i}(k_i')]]] > 0 \; (=)$$
$$r(\tau_i g(\hat{\theta}_{j \in \mathcal{N}(i)}) + (1-\tau_i)g(\alpha_{i,j \in \mathcal{N}(i)})) > c$$

## B. Gradient decomposition

We wish to maximize social welfare $U_{\text{Pop}} = B \sum_{t=1}^T \sum_{i=1}^N (\mathbb{1}[\frac{k_i}{M_i} \geq T] * r - a_{i,t} * c)$. Note that $a_i = \mathbb{1}[\pi_{C_i} - \pi_{D_i} > 0]$, meaning there are 2 step-functions $\mathbb{1}[\cdot]$ in $U_{\text{Pop}}$ where gradient is zero. Both can be replaced by sigmoids $\sigma(\cdot)$, leading to a differentiable approximation $\hat{U}_{\text{Pop}}$. We first analyse a further simplified $\hat{U}_C$, where the goal is to maximize the total number of cooperators in the population.

$U_C = \sum_{t=1}^T \sum_{i=1}^N a_{i,t} = \sum_{t=1}^T \sum_{i=1}^N \mathbb{1}[\pi_{C_i,t} - \pi_{D_i,t} > 0]$

Let $\tilde{a}_{i,t} = \sigma(\pi_{C_i,t} - \pi_{D_i,t})$ and $\hat{U}_C = \sum_{t=1}^T \sum_{i=1}^N \tilde{a}_{i,t}$.

$\nabla_\phi \hat{U}_C = \sum_{t=1}^T \sum_{i=1}^N \nabla_\phi \tilde{a}_{i,t}$

$= \sum_{t=1}^T \sum_{i=1}^N \nabla_\phi \sigma(\underbrace{rB[\tau_{t,i}(\phi)g(\hat{\theta}_{j \in \mathcal{N}(i)}(\phi)) + (1-\tau_{t,i}(\phi))g(\alpha_{i,j \in \mathcal{N}(i)})] - cB}_{h_{t,i}(\phi)})$

$= \sum_{t=1}^T \sum_{i=1}^N \sigma(h_{t,i}(\phi))(1 - \sigma(h_{t,i}(\phi)))\nabla_\phi h_{t,i}(\phi)$

$= \sum_{t=1}^T \sum_{i=1}^N \underbrace{\tilde{a}_{i,t}(\phi)(1 - \tilde{a}_{i,t}(\phi))rB}_{\psi_{t,i}(\phi)} \nabla_\phi[\tau_{t,i}(\phi)g(\hat{\theta}_{j \in \mathcal{N}(i)}(\phi)) + (1-\tau_{t,i}(\phi))g(\alpha_{i,j \in \mathcal{N}(i)})]$

$= \sum_{t=1}^T \sum_{i=1}^N \psi_{t,i}(\phi)\nabla_\phi[\tau_{t,i}(\phi)(g(\hat{\theta}_{j \in \mathcal{N}(i)}(\phi)) - g(\alpha_{i,j \in \mathcal{N}(i)}))]$

$= \sum_{t=1}^T \sum_{i=1}^N \psi_{t,i}(\phi)\nabla_\phi[\tau_{t,i}(\phi)(g(f_{j \in \mathcal{N}(i)}(a_{1:N}^{t-1}(\phi); \phi)) - g(\alpha_{i,j \in \mathcal{N}(i)}))]$

$= \sum_{t=1}^T \sum_{i=1}^N \psi_{t,i}(\phi)[(g(f_{j \in \mathcal{N}(i)}(a_{1:N}^{t-1}(\phi); \phi)) - g(\alpha_{i,j \in \mathcal{N}(i)}))\underbrace{\nabla_\phi \tau_{t,i}(\phi)}_{\text{accuracy}} + \tau_{t,i}(\phi)\underbrace{\nabla_\phi g(f_{j \in \mathcal{N}(i)}(a_{1:N}^{t-1}(\phi); \phi))}_{\text{steering}}]$

$\nabla_\phi \tau_{t,i}(\phi)$ can be interpreted as an accuracy component, where we are interested in having predictions that match past observations in order to increase trust. Interestingly, if the difference $g(f_{j \in \mathcal{N}(i)}(Y_{1:N}^{t-1}(\phi); \phi)) - g(\alpha_{i,j \in \mathcal{N}(i)})$ becomes negative, it means the model's current predictions are less cooperation-inducing than the agent's innate behaviour. In this case, the gradient will push to decrease accuracy, to incentivize agents to ignore predictions and instead follow their innate behaviour.

The second gradient $\nabla_\phi g(\hat{\theta}_{j \in \mathcal{N}(i)}(\phi))$, or equivalently $\nabla_\phi g(f_{j \in \mathcal{N}(i)}(Y_{1:N}^{t-1}(\phi); \phi))$, can be interpreted as a steering component. If trust $\tau_{t,i}(\phi)$ approaches zero, we won't care about steering since the agents are currently ignoring predictions.

The role of $\psi_{t,i}(\phi)$ is to scale the gradient. Gradients have a larger magnitude when $h_{t,i}(\phi)$ is close to zero, where the agent $i$ is closer to flipping her action between cooperate and defect.

Now let success $S_{i,t} = \mathbb{1}[\frac{k_{i,t}}{M_i} \geq T]$, its differentiable version $\tilde{S}_{i,t} = \sigma(\frac{k_{i,t}}{M_i} - T)$ and $\hat{U}_{\text{Pop}} = B \sum_{t=1}^{T} \sum_{i=1}^{N} (\tilde{S}_{i,t} * r - \tilde{a}_{i,t} * c)$.

$\nabla_\phi \hat{U}_{\text{Pop}} = B \sum_{t=1}^{T} \sum_{i=1}^{N} (r * \nabla_\phi \tilde{S}_{i,t} - c * \nabla_\phi \tilde{a}_{t,i})$

$\nabla_\phi \tilde{S}_{i,t} = \tilde{S}_{i,t}(1 - \tilde{S}_{i,t})\nabla_\phi(\frac{k_{i,t}}{M_i} - T) = \tilde{S}_{i,t}(1 - \tilde{S}_{i,t})\frac{1}{M_i}\nabla_\phi k_{i,t} = \tilde{S}_{i,t}(1 - \tilde{S}_{i,t})\frac{1}{M_i} \sum_{j \in \mathcal{N}(i) \cup \{i\}} \nabla_\phi \tilde{a}_{j,t}$

where each $\nabla_\phi \tilde{a}_{j,t}$ can be analyzed as in $\nabla_\phi \hat{U}_C$.

Optimizing for either $\hat{U}_{\text{Pop}}$ or $\hat{U}_C$ leads to qualitatively similar results empirically.

## C. Performance of architectures when maximizing accuracy

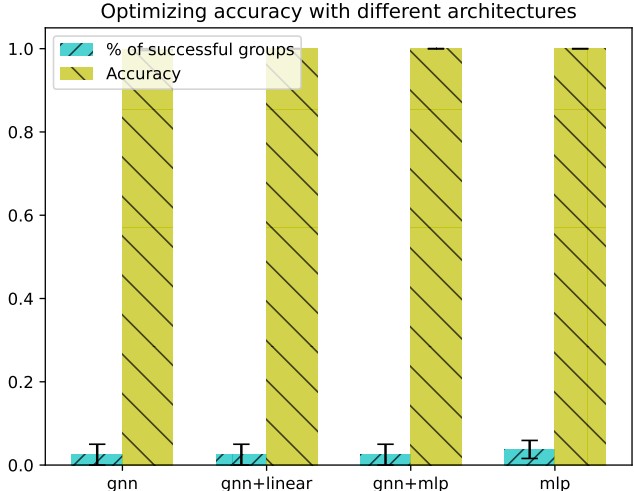

*Figure 5.* Performance of different architectures, when optimized to maximize social welfare. All parameters follow Figure 4.

## D. Visualizing a population playing CRD

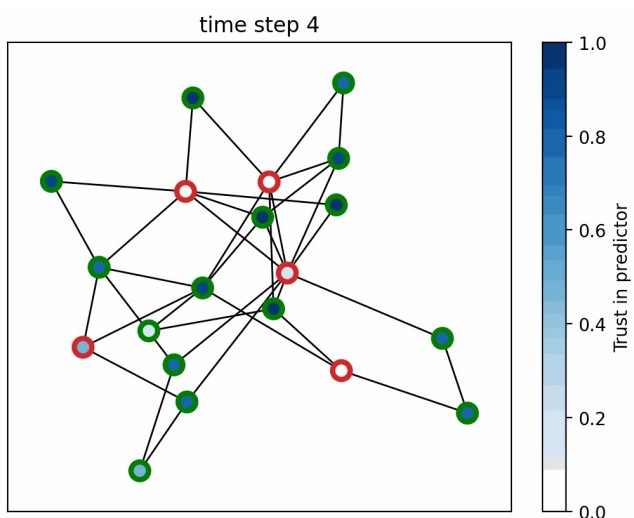

*Figure 6.* Population playing the Performative Collective Risk Dilemma over a scale-free network (Barabási & Albert, 1999). Circle borders indicate the agents' last action (green for cooperate, red for defect), and the filling indicates how much the agent currently trusts the predictor.

## E. Connections to existing frameworks

Unlike with the repeated risk minimization (RRM) algorithm from performative prediction, most work on strategic classification assumes knowledge of how the predicted adapt to a prediction. As a first step, we also assume this knowledge in our optimization procedure. Interestingly, we would not be able to apply RRM in our setting. This is because, unlike with accuracy, there is no gradient for welfare which does not flow through the agent adaptation (known as a mapping $\mathcal{D}(\theta)$ in Perdomo et al. (2020)). As a next step, one could assume a family of behaviours and estimate the correct one, as done in

Miller et al. (2021); Izzo et al. (2021).

## F. Full success is not always achievable

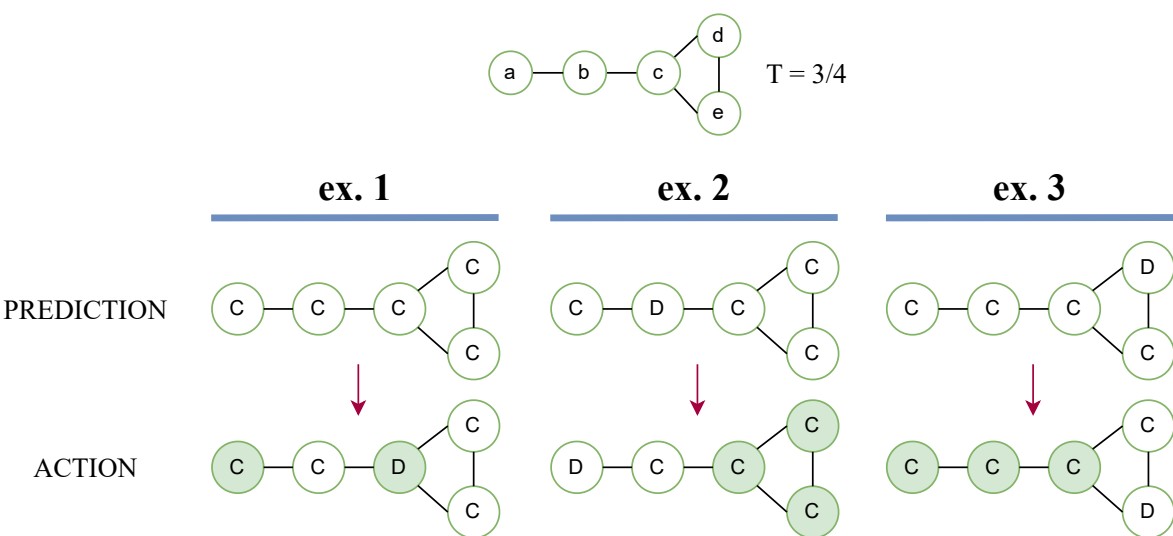

*Figure 7.* Achieving full success is not possible for all configurations of $\mathcal{G}$ and $T$. In this counter-example, node $c$ requires $\lceil TM_c \rceil = 3$ cooperators out of $M_c = 4$, meaning one node in $c$'s group can defect without preventing success. As a consequence $c$ will cooperate only if one of $b, d, e$ is predicted to defect. All the other groups require 100% of cooperators since they have $M_i < 4$ and $T = \frac{3}{4}$. If $c$ doesn't cooperate (ex. 1), it'll prevent success for its neighbours. If any of $b, d, e$ is predicted to defect (ex. 2 and 3), it'll also prevent someone's success. These contradicting requirements make it impossible to reach full success for any given prediction $\hat{\theta}$.

There exist combinations of $T$ and $\mathcal{G}$ for which full success is unattainable, even without requiring a self-fulfilling prophecy. This is due to contradicting requirements in neighbour nodes, which cannot be simultaneously satisfied through any prediction $\hat{\theta}$. One example is Figure 7.

A sufficient condition for full success to be unattainable is the following:

1. graph $\mathcal{G}$ has a "hub" node $H$ whose degree $M_H - 1$ is higher than any of its neighbours: $\forall i \in \mathcal{N}(H) : M_i < M_H$.

2. Threshold $T \in [0, 1]$ is set to $\frac{M_H - 1}{M_H}$.

3. $\forall i \in \mathcal{N}(H), \exists j \in \mathcal{N}(i) : M_j < M_H$.

With condition 2, for $H$ to overcome threshold, one out of $M_H$ agents does not need to cooperate (since $M_k \in \mathbb{N}$ and $\lceil TM_H \rceil = \lceil \frac{M_H - 1}{M_H} M_H \rceil = M_H - 1$). However, all neighbours $i \in \mathcal{N}(H)$ require 100% cooperators since they have $M_i < M_H \implies \lceil TM_i \rceil = M_i$. Condition 3 ensures each neighbour of $H$ is connected to another neighbour $j$ with low degree $M_j < M_H$. This combination requires all $i \in \mathcal{N}(H)$ to be predicted to cooperate (i.e. $\forall i \in \mathcal{N}(H), \hat{\theta}_i = 1$), otherwise their neighbours $j \in \mathcal{N}(i)/\{H\}$ will not cooperate (since they require 100% cooperators). However, $\forall i \in \mathcal{N}(H), \hat{\theta}_i = 1 \implies a_H = 0$ since $H$ can afford one defector in its group. Since $a_H = 1$ is a requirement for the success of $i \in \mathcal{N}(H)$, but that requires $\exists! i \in \mathcal{N}(H) : \hat{\theta}_i = 0$, we arrive at contradicting requirements.

This condition matches empirical observations in Figure 3. Thresholds that are close to but below 100% yield low success, even when maximizing welfare. This indicates that there may be no configuration which allows for high success, for settings $(\mathcal{G}, T)$ with high $T$.

Other counter-examples may be derived from this sufficient condition, such as those in Figure 8.

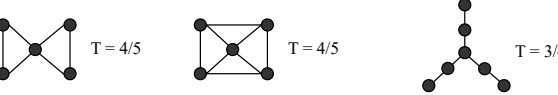

*Figure 8.* Other counter-examples where full success is not attainable.

