# OpenReview forum: "Performative Prediction on Games and Mechanism Design"
_ICML.cc/2024/Workshop/Agentic_Markets — Agentic Markets @ ICML'24 Poster_

### Official Review · Reviewer_VE9J · 2024-06-14
**Performative Prediction on Games and Mechanism Design - Review**

**Rating:** 6
**Confidence:** 3

**Review:**

The paper studies a game-theoretic model for predicting collective behavior. The authors consider a public good games that consist of a graph $G(V,E)$ with the vertices denoting the different agents. The action of each agent is either to cooperate or defect. If the number of defectors in an agents neighbors exceeds a certain threshold then agent incurs a cost however cooperating comes wtih an additional cost. Thus each agent decides to cooperate or defect based on estimate on the number of its neighbors that will cooperate or defect. This estimate comes as a global estimate of the agents behavior that agent $i$ can decide to trust or not. The authors addiitonally consider a bayesian update rule that the agent can use to update their trust on the global estimate of the agents' behaivor.

The author investigate under what condition the resultung multiagent system can converge to system success i.e. at each local neighborhood the set of agents cooperating always exceeds the required threshold. The authors provide a theoretical characterization in case on no bayesion updates in the agents' thrust as well as interesting experimental evaluation based on neural nets.

I overall believe that this is an interesting paper that should be considered for acceptance in the workshop.